# Microglial cyclooxygenase-1 modulates cerebral capillary basal tone in vivo in mice

William A. Mills III[1,2,3,4] ✉, Niesha A. Savory[1,2], Aida Oryza Lopez-Ortiz[1,2,5], Dennis H. Lentferink [1,2], Fernando González Ibáñez [6,7], Praise Agochi[1,2], Elina Rastegar[1,2], Arnav Gupta[1,2], Deetya Gupta[1,2], Arya Suram[1,2], Brant E. Isakson[3,8], Marie-Ève Tremblay [6,7,9,10,11,12,13] & Ukpong B. Eyo [1,2,3,4,5] ✉

Microglia and border associated macrophages have been implicated in hypercapnia, but it is unknown which myeloid cell modulates which vessel type. Previously, we documented in mice myeloid cell association with the brain vasculature but did not distinguish their localization along the vascular tree. Using molecular approaches to distinguish microglia and perivascular macrophages, we show that microglia are the only myeloid cells associating with capillaries. To determine if loss of microglia is sufficient to reduce capillary tone, we employ global and focal ablations and find significant reductions in capillary diameter and red blood cell flux, suggesting vasodilatory regulation by microglia. Cyclooxygenase-1 (COX1), an enzyme with known vasodilatory action, is predominantly expressed by microglia. To determine the necessity of microglial COX1 in regulating cerebral basal capillary tone in vivo, we perform genetic ablation and find a significant reduction in capillary flux and diameter. Together, this study using male mouse models reveals a role for microglial COX1 in maintaining basal capillary tone in vivo.

Although only weighing 2% of total body weight, the brain consumes 20% of total body energy[1,2]. This high energy demand relies on the delivery of oxygen and energy substrates via a dense vascular network[2,3]. Capillaries comprise the largest region of this network, reaching a length of 400 miles in human brain and possessing a surface area of ~20m$^2$ for molecular transport. These transport systems are essential for ensuring neuronal health and central nervous system (CNS) homeostasis. Thus, understanding mechanisms by which capillary blood flow is regulated in the normal brain could uncover potential therapeutic avenues to combat and/or prevent pathology known to have cerebral blood flow deficits, such as Alzheimer's Disease (AD)[4,5].

Microglial cells are yolk-sac derived CNS resident macrophages[6] with a unique repertoire of functions in the developing, mature and pathological brain[7–10]. We[11] and others[12] recently characterized a subset of microglia whose soma resides on capillaries, which we named capillary-associated microglia (CAMs). Depletion of these cells using the colony-stimulating factor-1 receptor (CSF1R) inhibitor, PLX3397, resulted in altered baseline CBF along with an impaired dilatory response upon $CO_2$ (hypercapnic) challenge as measured by laser-

[1]Department of Neuroscience, University of Virginia School of Medicine, Charlottesville, VA, USA. [2]Brain, Immunology, & Glia Center, University of Virginia School of Medicine, Charlottesville, VA, USA. [3]Robert M. Berne Cardiovascular Research Center, University of Virginia School of Medicine, Charlottesville, VA, USA. [4]Brain Institute, University of Virginia School of Medicine, Charlottesville, VA, USA. [5]Neuroscience Graduate Program, University of Virginia, Charlottesville, VA, USA. [6]Axe neurosciences, Centre de recherche du CHU de Québec-Université Laval, Québec, QC, Canada. [7]Division of Medical Sciences, University of Victoria, Victoria, BC, Canada. [8]Department of Molecular Physiology and Biological Physics, University of Virginia, Charlottesville, VA, USA. [9]Département de medicine moléculaire, Université Laval, Québec City, QC, Canada. [10]Department of Neurology and Neurosurgery, McGill University, Montréal, QC, Canada. [11]Department of Biochemistry and Molecular Biology, The University of British Columbia, Vancouver, BC, Canada. [12]Centre for Advanced Materials and Related Technology (CAMTEC), University of Victoria, Victoria, BC, Canada. [13]Insitute on Aging and Lifelong Health (IALH), University of Victoria, Victoria, BC, Canada. ✉e-mail: gne7xr@virginia.edu; ube9q@virginia.edu

speckle contrast imaging (LSCI)[11,13,14]. While we attributed these physiological results to microglia, the use of the PLX family of drugs (i.e., PLX3397 and PLX5622) has been shown to ablate other CSF1R-expressing myeloid cells, including perivascular and meningeal macrophages[15], which are collectively known as border-associated macrophages (BAMs)[16,17]. Moreover, LSCI, the imaging modality employed in recent studies[11,13,14], cannot sufficiently distinguish effects at different levels of the vascular tree. These confounds raise the question of whether (i) the observed vascular phenotypes are due to the loss of microglia, BAMs, or synergistic actions of both cell types and (ii) to what extent along the vascular tree myeloid cells contribute to vascular structure and function.

To address this limitation of previous work, using molecular approaches, we first determined that microglia largely reside at capillaries while perivascular macrophages reside at larger arterioles and venules. We then utilized longitudinal in vivo two-photon imaging through a cranial window to monitor microglial-capillary interactions. Employing global and focal ablation approaches revealed that loss of CAM results in significantly reduced capillary diameter and volume, respectively. Here, significant reductions in red cell flux could also be detected. Finally, given that the enzyme cyclo-oxygenase-1 (COX1) has been shown to mediate vasodilatory hypercapnic responses[18], our prior and current findings suggest that microglial COX1 might be necessary for maintaining basal capillary dilation. Following confirmation that the majority of COX1 in the brain resides in microglia, we utilized microglial-specific inducible genetic ablation to show that capillaries significantly constrict, and red blood cell flux significantly reduces following loss of microglial COX1, thus phenocopying the global and focal ablation results. Taken together, these studies point to microglia as the myeloid cell mediating basal capillary tone through the enzymatic action of COX1. Given unclear roles of COX1 in AD[19–21] and known capillary constrictions in pathology broadly[22–26], our findings raise the intriguing possibility for microglial COX1 as a promising therapeutic target in modulating the vascular deficits in AD specifically and brain pathology more broadly.

## Results

### Distinct localization between microglia and border-associated macrophages along the vascular tree

In our prior study[11], we showed that CX3CR1+ ramified myeloid cells localize to brain blood vessels. There, we used the CX3CR1-eGFP line in which eGFP is expressed in all myeloid cells, including perivascular macrophages and microglia. We described these cells as capillary-associated microglia (CAMs) but did not sufficiently confirm that all cells we counted as CAM were true microglia and not perivascular macrophages (PVMs) nor did we characterize myeloid cell-vascular associations along the vascular tree. To this end, we utilized the CX3CR1-eGFP mouse model[27] and immunohistochemistry to first confirm that all eGFP-expressing cell are Iba1+, a known marker specific for microglia and BAMs[28]. Results revealed that roughly 100% of eGFP+ cells also express Iba1 (Fig. 1a–e), matching prior findings[29,30]. While conducting this analysis, we additionally noticed one group of eGFP+ cells appeared to possess considerably higher fluorescent intensity than another group located at perivascular spaces. To confirm that microglia and PVMs possess differential eGFP fluorescent intensity, we immunostained for the microglia specific marker P2RY12 and PVM-specific marker CD206[31] in male CX3CR1-eGFP mice (Fig. 1f–k). We confirmed that indeed, P2RY12+:CD206-:eGFP+ microglia have on average, 50% greater eGFP fluorescent intensity relative to P2RY12-:CD206+:eGFP+ PVMs (Fig. 1l, Supplementary Fig. 1p). Additionally, these cells possess differential morphology, with microglia having a clear soma and ramified processes while perivascular macrophages are elongated and lack processes (Fig. 1j, k).

We then performed an analysis of eGFP^high intensity and CD206 area coverage (Supplementary Fig. 1a–j). Our results revealed that

CD206-:eGFP+ microglial volume is highest at capillaries (~65%), followed by venules (~19%) and α-smooth muscle actin+ arterioles (~16%; Fig. 1m–p). In contrast, CD206+:eGFP+ PVM volume was nearly double at arterioles (~64%) relative to venules (~35%), with minimal coverage at capillaries (~1%; Fig. 1m–p). To confirm our findings of CD206 area coverage, we then counted the number of NucBlueLive+ CX3CR1-eGFP^low PVMs at α-smooth muscle actin (αSMA) positive arterioles, αSMA negative venules, and capillaries <10 μm in size (Supplementary Fig. 1k, l). Results revealed that, while both arterioles and venules have PVMs (Supplementary Fig. 1l), they more frequently occupy perivascular space at arterioles. Hence, arterioles possessed the highest number of PVMs from our analysis relative to venules and capillaries (Supplementary Fig. 1m–o), matching our prior CD206 area analysis and confirming that PVMs do not occupy capillary territory. Taken together, microglia are the myeloid cell at higher branch-order upstream capillaries.

### Pharmacological elimination of myeloid cells and consequences to capillary basal tone

To determine how capillary basal tone would be impacted following global myeloid cell loss, we first wanted to confirm prior findings[15] that PLX3397 (herein referred to as PLX), ablates CD206+ meningeal macrophages and PVMs, as well as characterize their repopulation kinetics following withdrawal of PLX3397. Three cohorts of male mice (Fig. 2a) were treated with PLX at 660 mg/kg. At 8 days of PLX administration, CD206+ volume (Fig. 2b–g, k) including both CD206+ perivascular (Fig. 2b–g, m) and meningeal macrophages (Fig. 2b–g, o) were lost, along with CD206- microglia (Fig. 2b–g, q). However, only microglia successfully repopulate to baseline levels following 8 days of withdrawal from PLX administration and return to regular chow (Fig. 2b–j, q).

Longitudinal in vivo two-photon imaging in male CX3CR1-eGFP mice was used to assess the reconstitution of microglia following PLX withdrawal. Male mice underwent cranial window surgery and were allowed to recover for 14 days. In vivo imaging through a cranial window showed that microglia area is not changed at 7–14 days post-surgery relative to baseline images acquired immediately after surgery (Supplementary Fig. 2). With this paradigm, attempts were made to ascertain whether capillary "hotspots" exist for microglial association following their depletion. Examination of the same capillaries before and after microglial depletion/repopulation showed that ~a third of repopulated microglia returned to a spot that is within 5 μm of a CAM location prior to microglial elimination (Supplementary Fig. 3). Together, these results indicate different repopulation kinetics for microglia compared to BAMs following withdrawal from PLX with the possibility of some vascular "hotspots" that may attract microglia at capillaries.

With depletion and repopulation kinetics now known, we next sought to determine how 8 days of PLX administration would impact capillaries with CAMs compared to capillaries without CAMs. Interestingly, ~78% of capillaries that lost CAMs at 8 days of PLX administration had significantly reduced diameter (Fig. 3a–c, e–g), whereas ~22% had significantly increased diameter (Fig. 3a, b, d). Regardless of these differing changes in diameter, capillaries returned to baseline values at 8 days of PLX withdrawal when microglial repopulation had occurred (Figs. 3e–g and 2i,q). A similar trend in the phenotype was observed for those capillaries without CAMs, albeit diameter changes were not significant (Fig. 3h–k, l–n). These results suggest that microglia in male mice perform largely vasodilatory functions in the basal state with stronger effects at capillary locations with CAMs than capillary locations without CAMs.

Next, we examined possible effects of this depletion and repopulation approach on red blood cell flux using line scans of the same vessels at baseline, 8 days of PLX administration and at 8 days of PLX withdrawal when microglia have repopulated at capillaries (Fig. 4a).

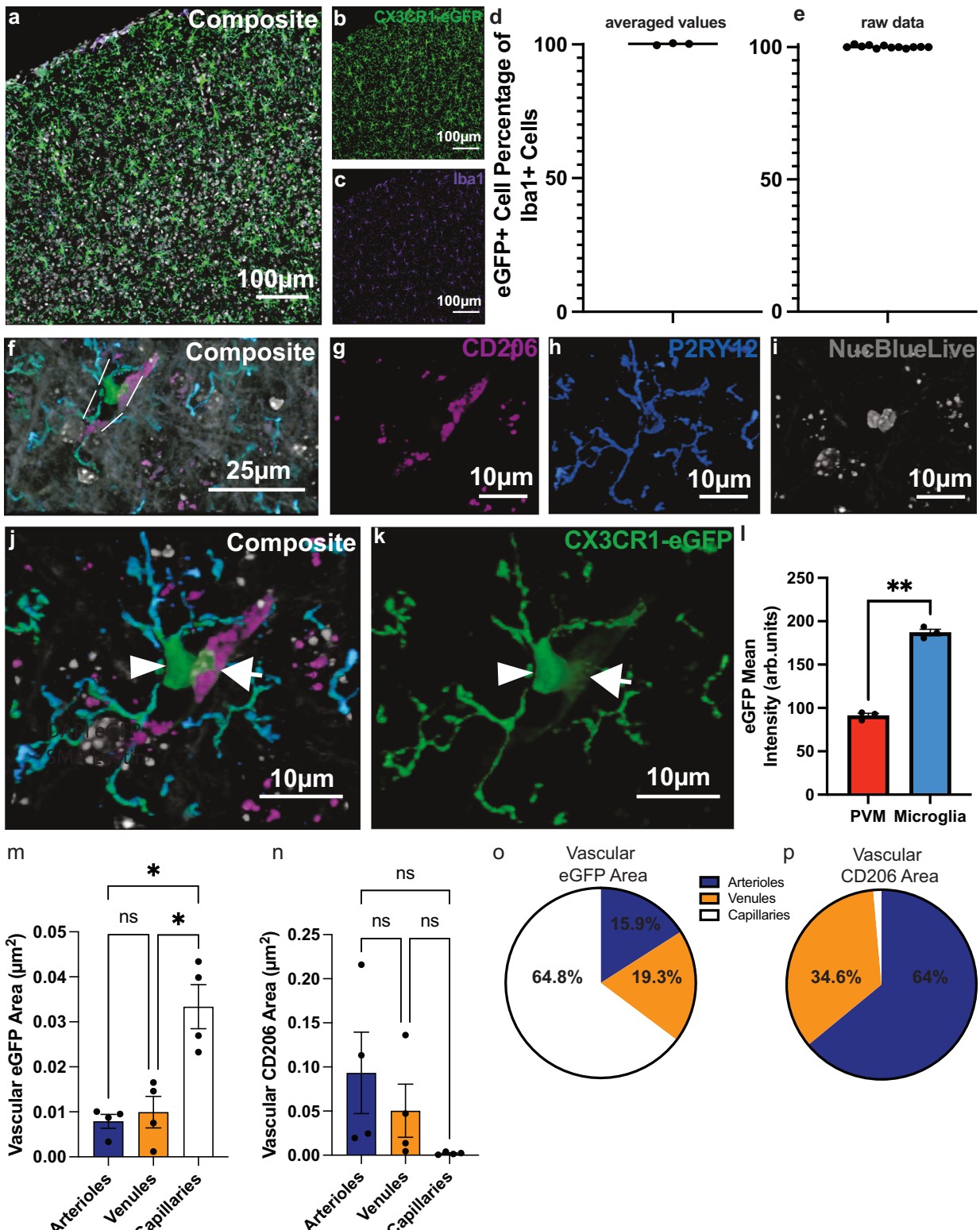

Consistent with the capillary diameter results (Fig. 3), we observed that red blood cell flux was significantly reduced by ~50% by 8 days of PLX with at least a partial recovery by 8 days of PLX withdrawal (Fig. 4b, c) in male mice. A few vessels show significant increases with PLX and partial recovery upon its withdrawal (Fig. 4d). However, this behavior was detected in only about 25% of capillaries analyzed while 75% of capillaries analyzed showed reduced blood cell flux (Fig. 4b, c, e). These results confirm vasodilatory roles for microglia in the

maintenance of capillary tone (diameter and blood flow). Microglia interact physically with pericytes[12,13], which may be an avenue by which they regulate capillary structure. We confirmed using electron microscopy that microglia can be seen juxtaposed to pericytes and endothelial cells at the ultrastructural level (Supplementary Fig. 4a). Similarly, immunohistochemistry for CD13 in male CX3CR1-eGFP mice showed juxtaposed microglia and pericytes along lectin-labelled capillaries (Supplementary Fig. 4b). This was also confirmed using

**Fig. 1 | P2RY12$^+$/CD206$^-$/CX3CR1-eGFP$^{high}$ ramified microglia are the myeloid cell localized to capillaries. a–c** 3D volumetric reconstructions from a 20X confocal image of somatosensory cortex in CX3CR1-eGFP mice, where **a** is the composite image, **b** the CX3CR1-eGFP channel, and **c** the Iba1 channel (purple). **d** Scatter dot plot showing averaged values of the percentage of CX3CR1-eGFP$^+$ cells that are also Iba1$^+$. $n = 12$ values across 3 mice. **e** Scatter dot plot showing the raw data used to generate the graph in (**d**). These 12 values represent 1 brain slice, 4 slices per mouse, across 3 mice. **f** 3D volumetric reconstructions from a 63X confocal image showing a microglial cell next to a perivascular macrophage (PVM). White lines indicate a blood vessel track showing the perivascular location of these cells. 3D volumetric reconstructions from a 63X confocal image showing the CD206 channel (magenta) in (**g**), the P2RY12 channel (blue) in (**h**), the nuclear label (grey) in **i**, the composite in (**j**), and the eGFP channel in (**k**). The arrowhead denotes the CX3CR1-eGFP$^{high}$, P2RY12$^{positive}$, CD206$^{negative}$ microglia. The arrow denotes the CX3CR1-eGFP$^{low}$, P2RY12$^{negative}$, CD206$^+$ PVM. **l** Bar graph quantifying the mean CX3CR1-eGFP

intensity of PVMs and microglia. $n = 67$ cells averaged across 3 mice. Two-tailed, paired $t$-test, $p < 0.0027$. **m** Bar graph quantifying the vascular coverage of eGF-Phigh microglia along specific vessel types. Brown-Forsythe and Welch ANOVA tests with Dunnett's T3 multiple comparison test. Multiplicity-adjusted $p$ values include arteriole vs. venule, $p > 0.9286$; arteriole vs. capillary, $p < 0.0194$; venule vs. capillary, $p < 0.0301$. $n = 43$, 20× somatosensory cortex images/5 mice. **n** Bar graph quantifying vascular coverage of CD206+ cells along specific vessel types. Brown-Forsythe and Welch ANOVA tests with Dunnett's T3 multiple comparison test. Multiplicity-adjusted p values include arteriole vs. venule, $p < 0.8215$; arteriole vs. capillary, $p < 0.3081$; venule vs. capillary, $p < 0.4267$. $n = 43$, 20X somatosensory cortex images/5 mice. For **m**, **n**, area coverage was normalized to total area of each vessel type per image. **o**, **p** Pie charts depicting the percentage distribution of CX3CR1-eGFP$^+$ microglia or CD206$^+$ border-associated macrophages along arterioles, venules, and capillaries, respectively. Data presented as mean ± s.e.m. LUTs have been adjusted in shown images. Source data are provided as a source data file.

double transgenic NG2-DsRed:CX3CR1-eGFP male mice[27,32] (Supplementary Fig. 4c).

## Focal elimination of microglia along capillaries leads to capillary constrictions

In combination with our vascular distribution study, the above PLX results suggest that CAMs specifically contribute to basal capillary tone. However, PLX is a global approach that affects all CSF1R-expressing cells, both within and outside the brain[33]. To interrogate more selectively microglial contributions at capillaries, we employed the focal 2Phatal ablation approach[34]. Based off our experience with 2Phatal from a prior study[35], we chose day 10 post-surgery to 2Phatal ablate CAMs, as ablation attempts are largely unsuccessful following this timepoint (Fig. 5a). 2Phatal ablation of CAMs resulted in significant reductions to capillary volume by 8 days post-ablation when there was no replenishment of microglia (Fig. 5b–g, h, i, l, m). Analysis at capillary regions adjacent to ablated regions (boxed regions in Fig. 5b, e) showed no change in capillary volume (Fig. 5j, k, n, o), further revealing the focal nature of this approach. Both failed ablation attempts (Supplementary Fig. 5a–f) and control ablations at parenchymal locations near the vessels but not targeting any microglia (Supplementary Fig. 5g–n) resulted in no change to capillary volume. Together, these focal microglial ablation studies along capillaries in male mice are consistent with the global elimination approach suggesting that microglia facilitate optimal basal capillary tone.

## Microglial COX1 regulates basal capillary tone

Given that both ablation approaches resulted in impaired capillary structure and flow, this suggests that microglia release vasodilatory molecules to mediate basal capillary tone. Cyclooxygenases (COX) are enzymes responsible for the production of eicosanoids from arachidonic acid (AA) by converting AA to prostaglandin $H_2$ (PGH$_2$)[36]. Terminal synthases catalyze PGH$_2$ conversion to other prostanoids such as prostaglandin $I_2$ (PGI$_2$), $E_2$ (PGE$_2$), $F_2\alpha$ (PGF$_2\alpha$), $D_2$ (PGD$_2$), and thromboxane $A_2$ (TXA$_2$), with each prostanoid or thromboxane exerting vasodilatory or contractile effects. Two isoforms of COX exist, COX1 and COX2, both of which are constitutively expressed in brain[37,38]. COX2 expression is localized to neurons in homeostatic conditions[38]. COX1 has recently been studied in astrocytes[39–41] although expression in microglia is acknowledged[42].

To begin to characterize COX1 expression in microglia, we first mined publicly available RNAseq databases[43–46]. Results show that microglia have the highest transcriptional expression of COX1 (Ptgs1), even higher than astrocytes (Supplementary Fig. 6). To determine if this was also true at the protein level, we performed immunohistochemistry in somatosensory cortex of male CX3CR1-eGFP mice. Quantification of COX1 in CX3CR1-eGFP$^{high}$:P2RY12$^+$ microglia (Fig. 6a–f), CX3CR1-eGFP$^{low}$:P2RY12$^-$ PVMs (Fig. 6a–f), and Aldh1l1-eGFP$^+$ astrocytes[47] (Fig. 6g–j) revealed that microglia possess at least

ten-fold higher expression of COX1 relative to PVMs and astrocytes (Fig. 6k, l). Notably, there is no difference in COX1 expression between CAM and non-CAM (Supplementary Fig. 7). These results suggest that microglia are the highest transcriptional expressors of Ptgs1 (COX1) and abundantly express COX1 protein to levels higher than astrocytes.

These findings were interesting given prior reports of astrocyte COX1 in the regulation of cerebrovascular physiology[39,41]. However, upon further examination, these studies were performed without cell-specific genetic ablation experiments. In light of this and our COX1 protein characterization, we utilized the TMEM119$^{creERT2}$ x Ptgs1$^{fl/fl}$ mouse model to test the necessity of microglial COX1 in the regulation of resting basal capillary tone. TMEM119$^{creERT2}$ was chosen as it will target microglia in brain without affecting perivascular macrophages[48]. Following implantation of a cranial window, male mice were allowed to recover for two weeks, upon which baseline z-stacks and line scan images were acquired. Tamoxifen was then administered once per day for five consecutive days and follow-up imaging performed 8 days following the last tamoxifen injection in accordance with our prior ablation timepoints (Fig. 7a). This resulted in a >50% reduction of COX1 in microglia of TMEM119creERT2 × Ptgs1$^{fl/fl}$ mice relative to TMEM119creERT2 x Ptgs1$^{fl/wt}$ mice by 8 days following the last tamoxifen injection, thus indicating effective COX1 reduction, albeit with a low n of 3 mice (Supplementary Fig. 8a–h). Assessments of capillary size and blood flow revealed a significant reduction in capillary diameter (Fig. 7b, f, h) and red blood cell flux eight days following the last tamoxifen injection (Fig. 7c, j). This was not the case for TMEM119$^{creERT2}$ x Ptgs1$^{fl/wt}$ (Fig. 7d, e, g, i, k). Notably, we observed no change in arteriole diameter in the same mice. (Supplementary Fig. 9). Taken together, these results demonstrate that microglial COX1 is necessary for resting basal capillary tone.

## Discussion

In this study using male mice, we confirm that microglia largely localize to capillaries whereas perivascular macrophages (PVMs) localize to arterioles and venules consistent with previous findings[49,50]. This was confirmed through two analyses, one being CD206 area coverage at blood vessels and the other through cell counting using a CX3CR1-eGFP$^{high}$ versus CX3CR1-eGFP$^{low}$ classification for microglia and PVMs, respectively. This classification has been observed and used heretofore[30], and in our hands, was the only valid way to visualize a nucleated perivascular macrophage soma. It is possible that our cell counts are lower than the full perivascular macrophage population given a recent study showing that there are CX3CR1-eGFP$^-$ cells that are Lyve1$^+$[50]. Importantly, these cells are also CD206$^+$. Therefore, while our cell counts could be lower, the important conclusion of PVM vascular location remains valid and matches that group's findings (See Fig. 2a in their paper)[50]. In addition, our finding is consistent with previous work showing that microglia contact capillaries through gaps between adjacent astrocyte endfeet[51].

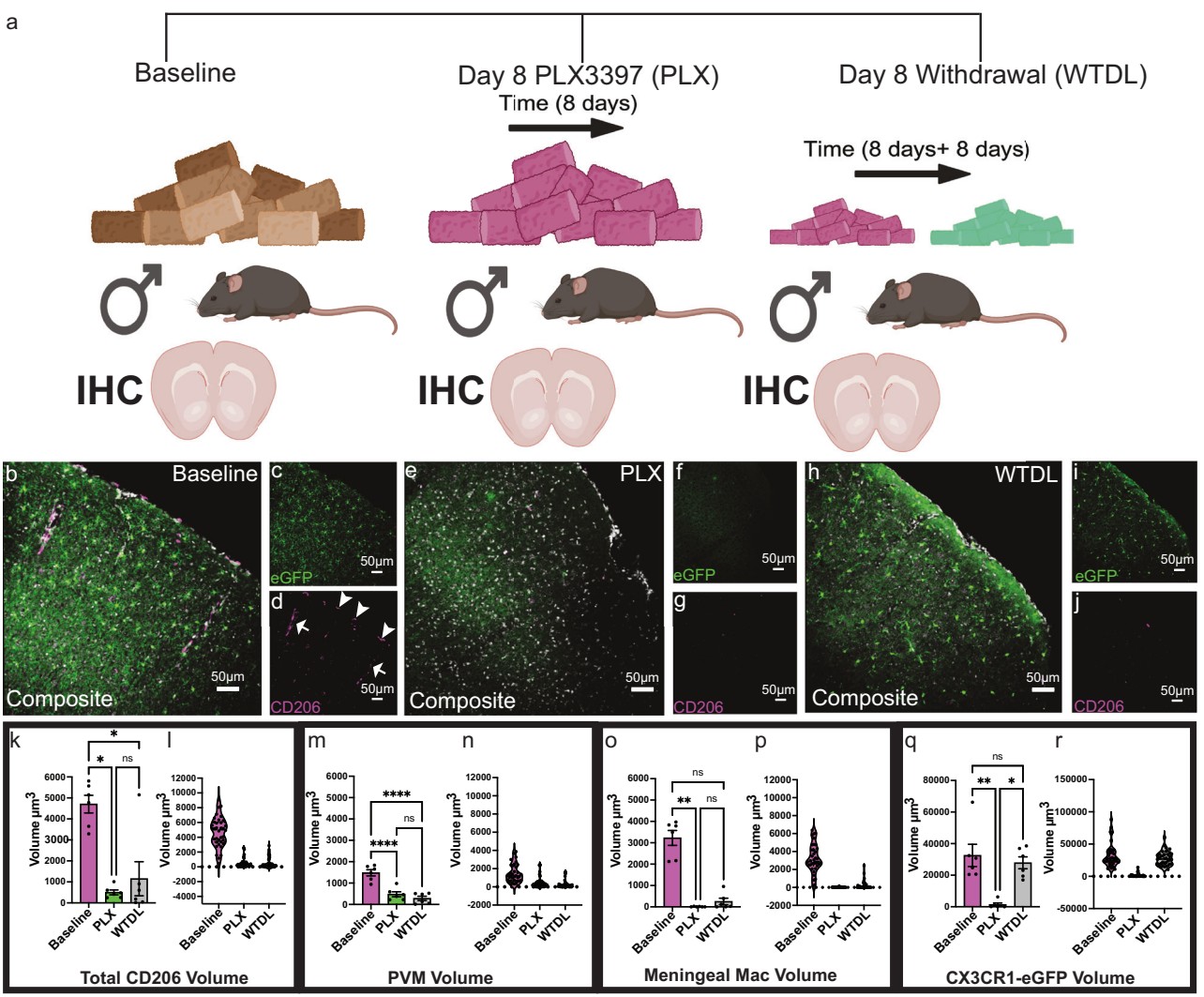

**Fig. 2 | PLX3397 ablates microglia and CD206+ meningeal and perivascular macrophages. a** Cartoon depicting the experimental approach (*Created in BioRender. Eyo, U. (2025)* https://BioRender.com/czwnlv4). **b** Representative composite 20X confocal image of somatosensory cortex in CX3CR1-eGFP mice for CD206 (magenta) and DAPI (grey), **c** the eGFP channel, and **d** the CD206 channel. Arrowheads denote the meningeal macrophages and arrows the perivascular macrophages. **e** The same as in b- except this composite representative image is from a mouse fed PLX3397 chow, **f** the eGFP channel, and **g** the CD206 channel. **h** The same as in **b** except this representative composite image is from a mouse that was fed control chow following PLX3397, **i** the eGFP channel, and **j** the CD206 channel. **k** Quantification showing averaged values from 42 fields of view/6 mice of total CD206 volume at baseline (mice fed control chow), mice fed PLX3397 (green), and repopulation (grey). Kruskal-Wallis test with Dunn's multiple comparisons test. Baseline vs PLX, *p* < 0.0305; Baseline vs. WTDL, *p* < 0.0205; PLX vs. WTDL, *p* > 0.09999. **l** Volcano plot of total CD206 volume raw data points. **m** Quantification showing averaged values from 42 fields of view/6 mice of total

perivascular macrophage volume at baseline (magenta), mice fed PLX3397 (green), and withdrawal (grey). One-way ANOVA with Tukey's multiple comparisons test. Baseline vs. PLX, *p* < 0.0001; Baseline vs. WTDL, *p* < 0.0001; PLX vs. WTDL, *p* < 0.5167 **n** Volcano plot of total CD206 PVM volume raw data points. **o** Quantification showing averaged values from 42 fields of view/6 mice of total meningeal macrophage volume at baseline, mice fed PLX3397, and withdrawal. Kruskal-Wallis test with Dunn's multiple comparisons test. Baseline vs. PLX, *p* < 0.0011; Baseline vs. WTDL, *p* < 0.0564; PLX vs. WTDL, *p* < 0.6880 **p** Volcano plot of raw data points. **q** Quantification showing averaged values from 42 fields of view/ 6 mice of total CX3CR1eGFP volume at baseline (mice fed control chow), mice fed PLX3397 (green), and withdrawal (grey). Kruskal–Wallis test with Dunn's multiple comparisons test, Baseline vs. PLX, *p* < 0.0074; Baseline vs. WTDL, *p* > 0.9999; PLX vs. WTDL, *p* < 0.0148 **r**) Volcano plot of raw data points. Data presented as mean ± s.e.m. LUTs have been adjusted in shown images. Source data are provided as a source data file.

The localization of microglia to capillaries is of high significance for vascular physiology since the capillary bed represents the location of highest nutrient exchange via the blood-brain barrier[52], and with few exceptions, a neuron is rarely more than 8–20 μm from a capillary[53]. Therefore, it is likely that microglia may be the best situated to regulate vascular function amongst myeloid cells for several reasons. First, they are more numerous than PVMs and therefore could have a more extensive effect. Second, microglia are ubiquitously and evenly distributed throughout the brain providing opportunities for broad and local control of blood flow. Finally, since they are localized to

capillaries, which are the vascular elements of closes proximity to neural cells, they are well-positioned to maintain regulatory control of vascular function.

To determine whether microglia could regulate capillary basal tone, we sought to interrogate whether microglial loss at capillaries could instruct capillary size and flux changes. We observed significantly reduced capillary diameter following global ablation of microglia, which was restored to baseline values following microglial repopulation. Further complementing these pharmacological findings, we provide evidence that focal elimination of microglia at capillaries

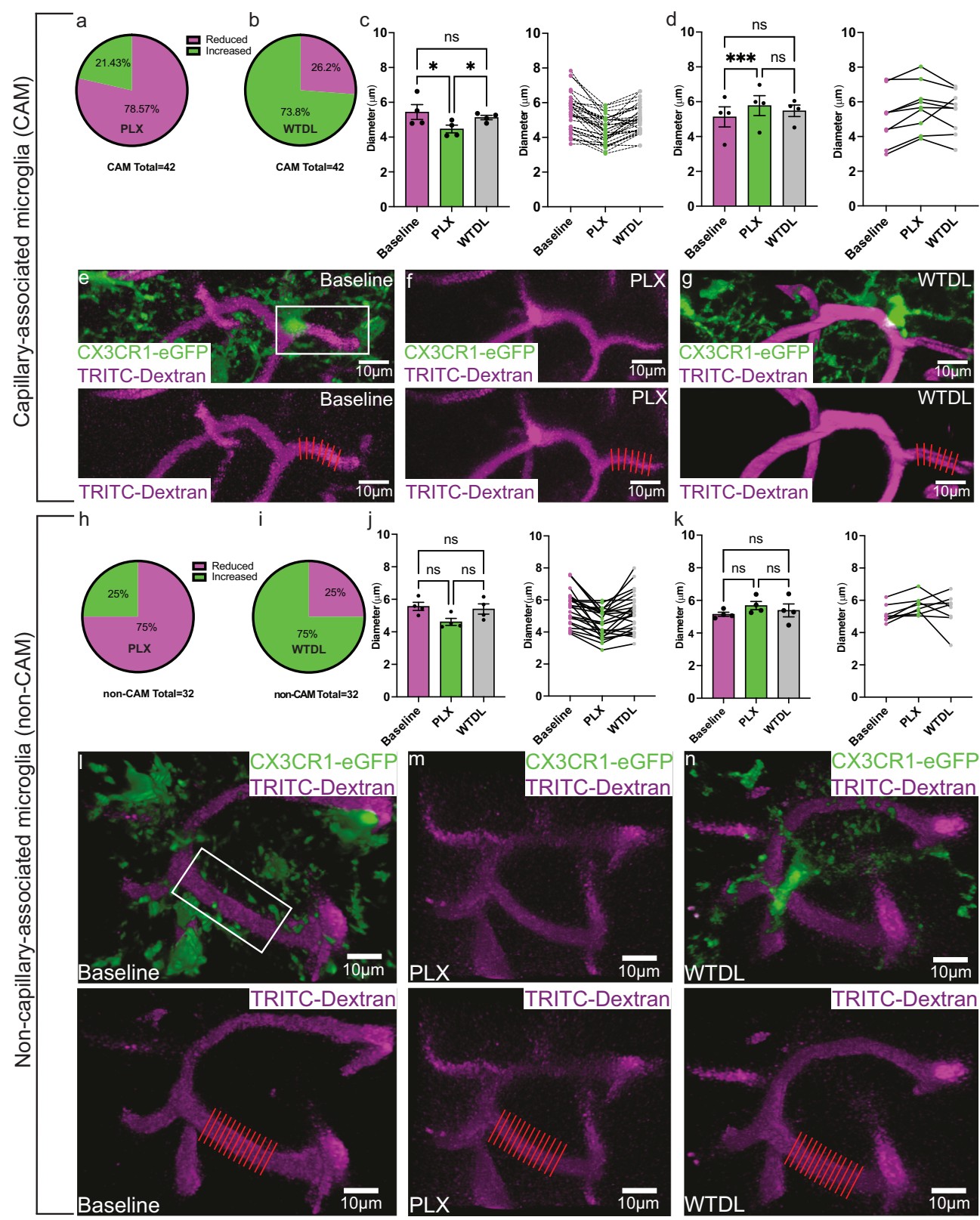

also elicits reductions in capillary size in a way that does not occur with failed ablations or non-microglial tissue ablations. Findings from these global and focal ablation studies mirror what Morris et al.[12] showed with longitudinal in vivo two-photon imaging where normally occurring loss of CAM resulted in significantly reduced capillary diameter (see Fig. 5i). Likewise, normally occurring gain of microglial interaction resulted in increased capillary diameter, albeit to nonsignificant levels (see Fig. 5h). Moreover, since focal astrocytic ablations did not alter capillary diameter[35], these results suggest a microglial-specific role in maintaining a dilated basal tone.

Given that the same trend in diameter reduction was observed for capillaries with or without CAM, albeit insignificant without CAM

**Fig. 3 | Effect of global ablation of myeloid cells on capillaries. a** Pie chart depicting the percentage of capillaries laden with CAM that increased or decreased in diameter by day 8 PLX. **b** Pie chart showing the percentage of capillaries laden with CAM that increased or decreased in diameter by day 8 repopulation (labeled WTDL). **c** (left) Bar graph comparing averaged diameter value reductions from 33 capillaries/4 mice. One-way ANOVA with Tukey's multiple comparison test. Baseline vs. PLX, $p < 0.0462$; Baseline vs. WTDL, $p < 0.6945$; PLX vs. WTDL, $p < 0.0376$. **c** (right) Before-and-after line plot showing raw values used to construct **c** (left). **d** (left) Bar graph comparing averaged diameter value increases from 9 capillaries/4 mice. One-Way ANOVA with Tukey's multiple comparison test. Baseline vs. PLX, $p < 0.0006$; Baseline vs. WTDL, $p < 0.4547$; PLX vs. WTDL, $p < 0.5434$. **d** (right) Before-and-after line plot showing raw values used to construct **d** (left). **e** 3D volumetric reconstruction where **e** is the composite image (top) with just the TRITC-Dextran channel (magenta) below. The white box in **e** indicates the measured CAM laden capillary portion. **f** 3D volumetric reconstructions showing the same capillary at day 8 PLX. **g** 3D volumetric reconstructions showing that same capillary at day 8 withdrawal (WTDL). The red dashed lines in **e**–**g** (bottom) represent the Vasometrics output used to generate the reported capillary diameter value. **h**–**n** The same as **a**–**g**, just for non-CAM laden capillaries. **j** (left) Bar graph comparing averaged diameter value reductions from 24 capillaries/4 mice. One-way ANOVA with Tukey's multiple comparison test. Baseline vs. PLX, $p < 0.0917$; Baseline vs. WTDL, $p < 0.9401$; PLX vs. WTDL, $p < 0.2001$. **j** (right) Before-and-after line plot showing raw values used to construct **j** (left). **k** (left) Bar graph comparing averaged diameter value increases from 8 capillaries/4 mice. One-Way ANOVA with Tukey's multiple comparison test. Baseline vs. PLX, $p < 0.0518$; Baseline vs. WTDL, $p < 0.8209$; PLX vs. WTDL, $p < 0.7578$. **k** (right) Before-and-after line plot showing raw values used to construct **k** (left). Data presented as mean ± s.e.m. LUTs have been adjusted in shown images. Source data are provided as a source data file.

(Fig. 3), we think this result suggests that CAM and non-CAM cells may perform similar functions at capillaries. RNAscope findings from our prior work[11] further suggest that CAM and non-CAM are not transcriptionally distinct microglia. Therefore, while abluminal positioning may not be necessary for vessel tone regulation, some sort of physical proximity is, as supported by both our 2Phatal and vascular hotspot data. We now show that microglia signal via the enzymatic action of cyclooxygenase-1 (COX1). Here, we show that they are the highest COX1-expressing cells in the brain, even more so than astrocytes and perivascular macrophages. Through the use of genetic approaches, we confirm that microglial specific loss of COX1 is sufficient to elicit reductions in capillary diameter and red blood cell flux, thus implicating microglial COX1 enzymatic action in the regulation of basal capillary tone. Indeed, since COX1 is expressed at similar levels in both CAM and non-CAM microglia, it is reasonable that both CAM-laden capillaries and capillaries without CAMs show similar trends in capillary size responses during global microglial elimination. These results suggest that all microglia are capable of regulating basal capillary tone, though this regulation is heightened by physical proximity.

Finally, studies[54] have shown increased capillary diameter and RBC flux following focal pericyte ablation. Our results combined with these studies suggest that microglia and pericytes may work in tandem to regulate capillary basal tone, and future studies will elucidate the prostanoid signaling downstream of COX1 by which microglia communicate with pericytes to modulate basal capillary tone. By better understanding the cellular and molecular mechanisms regulating CBF, new therapeutic avenues can be uncovered for treating blood flow deficits in conditions such as Alzheimer's[4,5].

## Limitations of our study

Prior studies have implicated astrocytic COX1 in neurovascular coupling[39,41]. We do not take our findings to question these prior findings for two notable reasons. First, these studies occur at the first branching segment from a penetrating arteriole, while our study focuses on higher-order capillaries. Second, astrocytes express COX1 based on our findings and hence can signal to underlying mural cells to mediate vascular physiology. In addition, given our microglia-specific targeting approach through use of the TMEM119 promoter, we cannot exclude a role for perivascular or meningeal macrophages in mediating basal capillary tone. Moreover, while with our in vivo imaging we identified microglia by their high expression of GFP under the CX3CR1 promoter, for our COX1 studies, we targeted microglia through TMEM119 inducible Cre expression. Consistent with past literature, we assume that these are the same populations of cells i.e. microglia. However, we do not directly show this.

## Methods

All studies were approved by the Institutional Animal Care and Use Committees of the University of Virginia (protocol number 4237) and were conducted in compliance with the National Institutes of Health's 'Guide for the Care and Use of Laboratory Animals'. For electron microscopy, animal protocol was approved by the University of Victoria's Animal Care Committee under the guidelines of the Canadian Council on Animal Care.

### Mice

Male mice aged 2–3 months were used for all studies, with studies in supplementary Fig. 3 being an exception. All mice were housed under controlled temperature, humidity, and light (12:12 h light-dark cycle) with food and water readily available ad libitum. No more than five mice were housed together in a single cage. CX3CR1-eGFP[27] and NG2-DsRed mice[32], TMEM119creERT2 (Jax Strain # 031820[55]), and Ptgs1 fl/fl (Jax Strain # 030884), mice were used for this study. Swiss Webster-Aldh1l1-eGFP bacterial artificial chromosome transgenic mice (generated by the GENSAT project) were a generous gift from Harald Sontheimer.

### Electron microscopy

For electron microscopy, mice were anaesthetized with rodent cocktail (0.3 mL/100 g) containing ketamine [100 mg/mL], xylazine [20 mg/mL] and aceprozamine [10 mg/mL] before perfusion with ~15 mL of ice-cold PBS, ~75 mL of 3.5% acrolein in PB and ~125 mL of 4% PFA. Brain was extracted, post-fixed for 2 h in 4% PFA, rinsed with PBS and cut into 50 μm thick section using a Leica VT1200S vibratome. Brain sections were then collected and stored in cryoprotectant solution at −20 °C until use. Sections containing the barrel cortex (Bregma −1.67 mm) were processed for immunocytochemistry staining for ionized calcium-binding adapter molecule 1 (IBA1) as performed in Bordeleau, 2022. Specifically, sections were washed in PBS, quenched for 10 min in 0.3% $H_2O_2$, washed, permeabilized in 0.1% $NaBH_4$ for 30 min, and washed again. Sections were subsequently placed in 10% fetal bovine serum, 3% bovine serum albumin, 0.01% Triton X-100 in 50 mM TBS, pH = 7.6 blocking buffer for one hour at room temperature followed by an overnight incubation with rabbit-anti-Iba1 (applied at 1:1000 in blocking buffer- Wako Chemicals Cat. No. 018-28523) at 4 °C. The following day, sections were washed in Tris-buffered saline (TBS), incubated at room temperature for 1.5 h with biotinylated goat-anti-rabbit secondary antibody (applied 1:300 in TBS; JacksonImmunoResearch cat # 111-066-046). This was followed by the avidin-biotin-complex applied at 1:1000 in TBS (Vector Laboratories, catalogue number PK-6100) for 1 h at room temperature, then washed and revealed in 0.05% diaminobenzidine (DAB, 0.015% $H_2O_2$) in TBS-Millipore Sigma catalogue number D5905-50TAB). To process for electron microscopy, sections were then incubated in 3% ferrocyanide (in H2O; cat# PFC232.250, BioShop, Burlington, ON, Canada) combined (1:1) with 4% aqueous osmium tetroxide (cat# 19170, Electron Microscopy Sciences, Hatfield, PA, United States) for 1 h, washed in PBS, incubated in 1% thiocarbohydrazide (in PBS; cat#

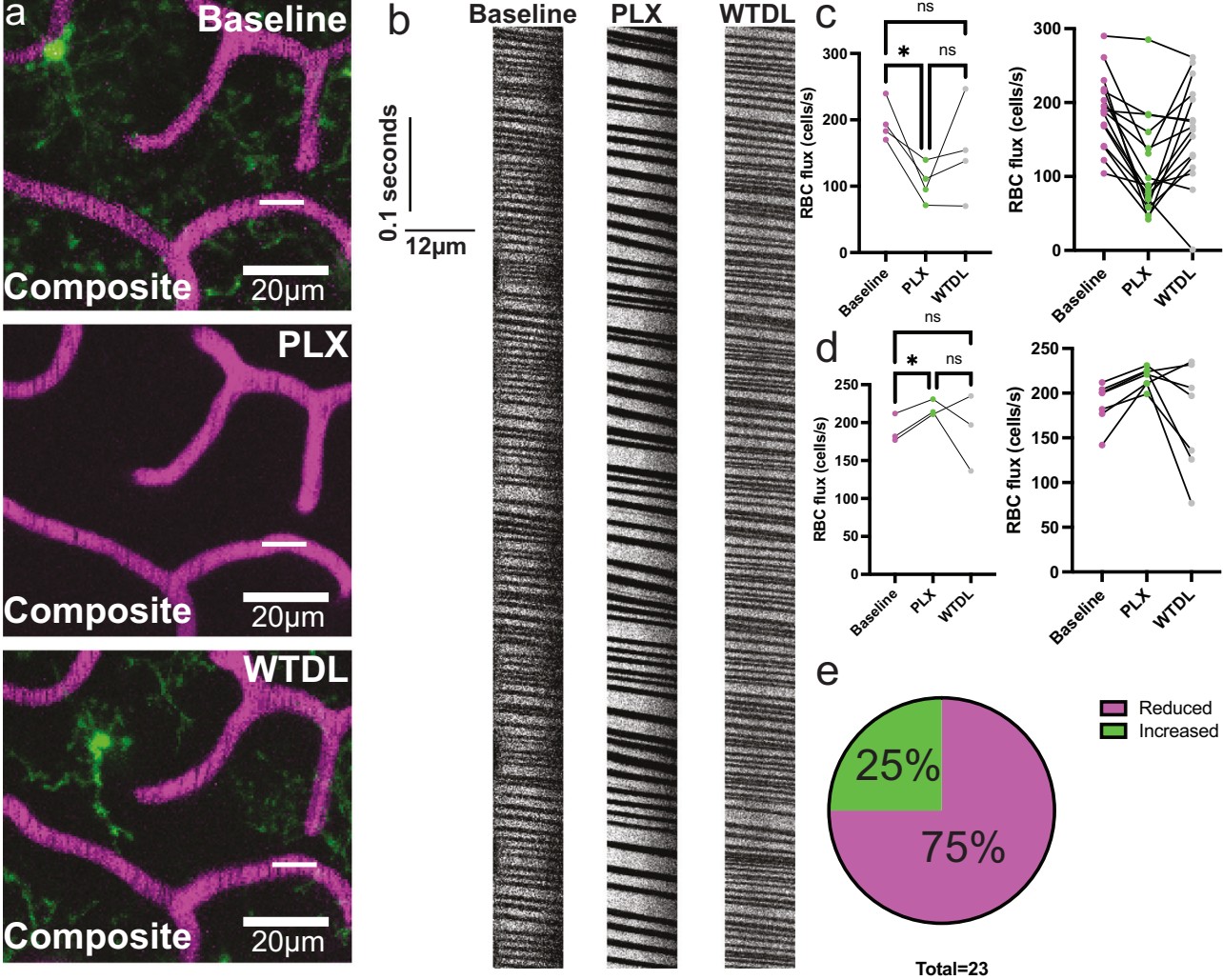

**Fig. 4 | Global microglial ablation using PLX3397 results in significantly reduced capillary red blood cell flux. a** Top image- maximum intensity projection showing a capillary (magenta) at baseline from which a line scan was acquired, as indicated by the white line. Middle image- maximum intensity projection showing that exact same capillary from baseline following 8 days of PLX3397 administration (PLX), from which a line scan was acquired again, as indicated by the white line. Note the absence of CX3CR1-eGFP+ cells. Bottom image- maximum intensity projection showing that exact same capillary from baseline but at 8 days following PLX3397 withdrawal (WTDL), from which a line scan was acquired again, as indicated by the white line. **b** Representative line scan kymographs from the generated line scans shown in **a**, where baseline is shown on the left, day 8 PLX in the middle, and day 8 WTDL on the right (labeled as baseline, PLX, WTDL- respectively). **c** Left, before-and-after plot showing averaged values of capillary red blood cell flux in capillaries that demonstrated reduced flux following 8 days of PLX3397

administration. $n = 16$ capillaries/4 mice, with all mice being in the same acquisition cohort. One-way ANOVA with Tukey's multiple comparisons test. Baseline vs. PLX, $p < 0.0436$; Baseline vs. WTDL, $p < 0.2684$; PLX vs. WTDL, $p < 0.4553$. Right, raw values used to generate the before-and-after plot on the left. **d** Left, before-and-after plot showing averaged values of capillary red blood cell flux in capillaries that demonstrated increased flux following 8 days of PLX3397 administration. $n = 7$ capillaries/3 mice. One-way ANOVA with Tukey's multiple comparisons test. Baseline vs. PLX, $p < 0.0479$; Baseline vs. WTDL, $p < 0.9995$; PLX vs. WTDL, $p < 0.6486$. Right, raw values used to generate the before-and-after plot on the left. **e** Pie chart showing that 75% of capillaries in this study demonstrated reduced flux following 8 days of PLX3397 administration (magenta), while 25% demonstrated increased flux following 8 days of PLX3397 administration (green). Source data are provided as a source data file.

2231-57-4, Electron Microscopy Sciences) for 20 min, washed in PBS, incubated in 2% osmium tetroxide (in H2O), then dehydrated in ascending concentration of ethanol (35%, 50%, 70%, 80%, 90%, and 3 times in 100%) followed by incubation in propylene oxide. Post-fixed sections were embedded in Durcupan ACM resin (cat# 44611−44614, MilliporeSigma) for 24 h, placed between two ACLAR® embedding sheets (cat# 50425-25, Electron Microscopy Sciences) and resin was polymerized at 55 °C for 72 h. The region of interest−barrel cortex−was excised, glued on a resin block, and cut into 75 nm thick ultrathin sections using a Leica Ultracut UC7 ultramicrotome (Leica Biosystems). Ultrathin processed sections were collected on a silicon nitride chip, glued on specimen mounts, and imaged at 5 nm resolution (x, y) using a Zeiss

Crossbeam 540 Gemini scanning electron microscope, operating with an acceleration voltage of 1.4 kV and current of 1.2 nA. Microglia were identified by their immunoreactivity to Iba1 and ultrastructural features. Capillaries were identified by the presence of a basal membrane surrounding the capillary cells: endothelial cells and pericytes.

**In vivo multiphoton imaging through a cranial window**
To perform cranial window surgery, induction of surgical plane anesthesia with 2–5% isoflurane was first established. Pre-operative analgesics (Bupivacaine) were then administered subcutaneously at the site of incision prior to surgery. Hair and skin of the skull was then subsequently removed, and a 3 mm craniectomy 2 mm posterior and

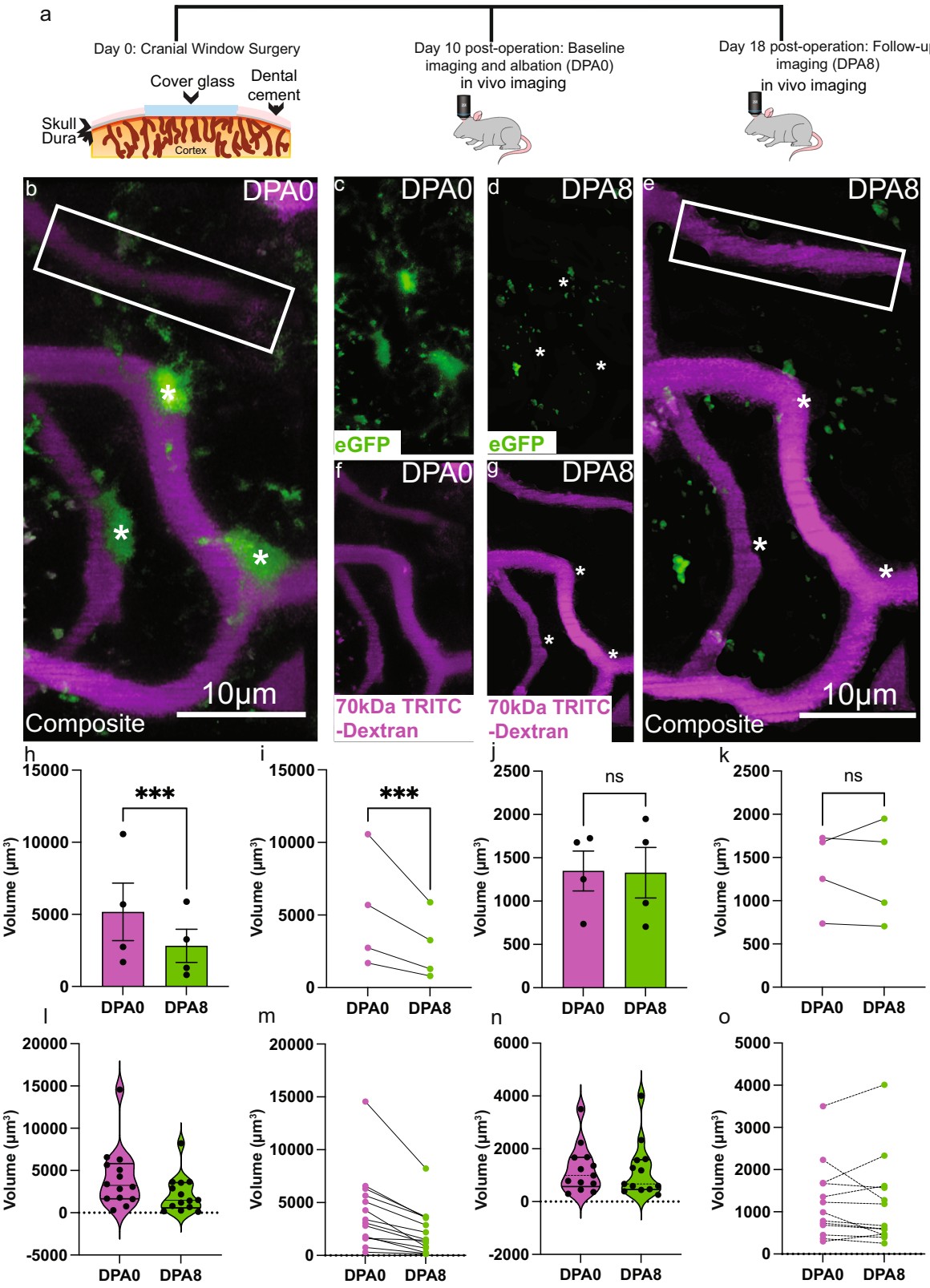

1.5 mm lateral to bregma performed on one hemisphere. Dura was removed next, followed by placement of a 3 mm circular No. 1 cover glass (Warner Instruments CS-3R, Cat #: 64-0720). This was glued to the skull using surgi-lock 2oc (Meridian Animal Health) plus affixed and sealed with C&B Metabond dental cement (ParkellRef S380). Buprenorphine SR was then administered intraperitoneally at the end of surgery. All mice were allowed to recover 14 days before experiments

commenced, with 2Phatal experiments being the exception (discussed below). To image, animals were placed on a Kopf stereotax with heating pad and were anaesthetized with Isofluorane to ~100 beats per minute, with their vitals constantly monitored. Cerebral vessels were visualized by retro-orbital injection of 70 kDa TRITC-Dextran (25 mg/mL, ThermoFisher catalog number D1818) and Rhodamine (2 mg/mL, ThermoFisher catalog number A-30677). Imaging was

**Fig. 5 | Focal 2Phatal ablation of capillary-associated microglia (CAM) results in significantly reduced capillary volume only in the CAM-laden region but not adjacent capillary stretches. a** Cartoon depicting the experimental approach. **b** 3D volumetric reconstruction in somatosensory cortex of CX3CR1-eGFP mice showing a stretch of capillary (magenta) where CAM were ablated, as indicated by asterisks. The rectangle indicates a region of capillary where CAM was not ablated, and volume measured at baseline, or day post-ablation 0 (DPA0). **c** 3D volumetric reconstruction of the same region and timepoint as in **b**, but just the eGFP channel. **d** 3D volumetric reconstruction of the same region as in **b**, but just the eGFP channel and at 8 days post-ablation (DPA8). **e** 3D volumetric reconstruction of the same region as in **b**, but at DPA8. Asterisks indicate the successfully ablated microglial cells, and the rectangle indicates a region of adjacent capillary where CAM was not ablated, and volume measured. **f** 3D volumetric reconstruction of the same region and timepoint as in **b**, but just the 70 kDa TRITC-Dextran channel. **g** 3D

volumetric reconstruction of the same region as in **b**, but just the 70 kDa TRTIC-Dextran channel at DPA8. **h** bar graph comparing the averaged values of capillary volume at baseline (DPA0) and following CAM ablation at DPA8. Two-tailed ratio paired $t$-test, $p < 0.0010$. $n = 14$ capillaries/4 mice. **i** Before-and-after line plot showing the same data in (**h**). **j** bar graph comparing the averaged values of capillary volume at locations adjacent to CAM-ablated regions at baseline (DPA0) and following CAM ablation at DPA8. Two-tailed ratio paired $t$-test, $p < 0.6391$. $n = 13$ capillaries/4 mice. **k** Before-and-after line plot showing the same data in (**j**). **l** Violin plot of the raw data used to derive the bar graph shown in (**h**). **m** Before-and-after line plot showing the same raw dataset as in (**l**). **n** Violin plot of the raw data used to derive the bar graph shown in (**j**). **o** Before-and-after line plot showing the same raw dataset in (**n**). Data presented as mean ± s.e.m. LUTs have been adjusted in shown images. Source data are provided as a source data file.

performed 100–200 μm below the surface of somatosensory cortex. For all data acquired, a zoom 1 overview was acquired to enable finding the exact same fields of view at later timepoints. A zoom 1.5 or zoom 2 image was then acquired and clearly annotated to further facilitate finding the exact same blood vessels at later timepoints in the experiment. For z stacks, a location used to set the laser power and optimize the histogram was selected and annotated in the zoom 1.5 or 2 images so that this exact same location could be utilized at later timepoints. For line scan imaging, breaths per minute (b.p.m.) was recorded at baseline and then a line applied to bisect the lumen of a higher-order capillary. Pixel dwell time was set to 2.0 μs for all line scan segments. At subsequent time points, isofluorane was adjusted until breath rate was within 10 b.p.m. of baseline values, and then the settings from the baseline image applied to capture a line scan from the same capillary following experimental intervention (Figs. 4 and 7). A Chameleon Vision II (Coherent) laser tuned to 870 nm was used to excite all dyes. Optical sections were acquired using an Olympus Dual beam FVMPE-RS multiphoton microscope, with Spectra Physics InSight X3 pulsed laser utilized for excitation. Two cooled GaAsP and two multialkali photomultiplier detectors allowed for simultaneous 4-channel multiplexing, and a XLPLN25X/1.05 NA water-immersion objective (Olympus) was used for image acquisition. Barrier filters used included Olympus FV30-FVG (Blue light at 410–455 nm and green light at 495–540 nm−dichroic at 475 nm) and FV30-FRCY5 (red light at 610/70 and far red at 705/90-dichroic at 650 nm). The FV30-SDM570 dichroic mirror was utilized to split light <570 nm to the RNDD3/4 detector and light >570 nm to the RNDD1/2 detectors. Z projections were created using FIJI (NIH) and Nikon Imaging Software (NIS)-Elements, with optical section thickness set to 1 μm.

## 2Phatal ablation

To induce cell death in individual microglia, mice underwent the surgical procedure described above, where following a durotomy, Hoechst 33342 (ThermoFisher cat # H5370) was applied topically (0.08 mg mg⁻¹ diluted in PBS) to the cortex of CX3CR1-eGFP mice over ten minutes- then washed thoroughly with cold 1× PBS. above Mice were then allowed to recover for ten days. To ablate single microglia, an ROI was placed over the nucleated soma of a microglia. Photobleaching was achieved by setting pixel dwell time to 100 μs/pixel and laser wavelength to 775 nm. Total scan time in the ROI was set to 20 s. Increased laser power was used with increasing depth and/or decreased Hoechst intensity, as measured in the activation ROI.

## Immunohistochemistry

To label the vasculature, CX3CR1-eGFP mice were retro-orbitally (RO) injected with stock concentration Lycopersicon Esculentum Dylight 594 (Vector Labs DL-1177). Mice were weighed, and blood volume was calculated as 7% of total body weight. No greater than 7% of total blood volume of Lectin 594 was RO injected. Two and-a-half hours post injection, mice were then injected with Ketamine/Xyalzine, where mice

used for CD206 staining were then subsequently transcardially perfused with PBS/heparin (5 U/mL) followed by 4% paraformaldehyde (PFA) in PBS. Brains were subsequently immersed in 4% PFA overnight. Mice used for COX1 and Aquaporin-4 staining only underwent exsanguination using PBS/heparin (5 U/mL) followed by drop-fixation in 4% PFA for 48 h. Note that Iba1 and P2RY12 staining occurred in studies using both fixation procedures. Brains from all groups were then serially sectioned into 30 μm sections on a Leica VT1000S vibratome and placed in a 24-well plate with PBS 0.01% Na⁺ Azide in each well. To select slices for immunohistochemistry, google random number generator was used to randomly select for wells 1–12 or 1–24. Six slices with somatosensory cortex were chosen from every 4th well. Four slices were used for primary and secondary staining, the 5th for a secondary only control, and the 6th for a tissue only control. In the original CD206 stain (Fig. 1m−p, Supplementary Fig. 1a–j), slices were blocked for 1 hour in 1% BSA 2% Triton X 100 in 1× PBS at room temperature on an orbital shaker. Sections were then washed 3 times in 0.1% Triton X 100 in 1× PBS, with each wash lasting 10 min. Primary antibodies CD206 (BioRad Cat # MCA2235, Lot # 148572) and α-smooth muscle actin (D4k9N) (Cell Signaling Technology Cat. No. 19245 Lot 6) were bath applied at 1:100 in blocking buffer overnight and placed on an orbital shaker at 4 °C. The next day, slices were washed 3 times for 10 min each in 0.1% Triton X 100 in 1× PBS at room temperature. Secondary antibodies were then applied at 1:500 in the same solution for 1 h, where goat-anti-rat 647 (Invitrogen Cat # A48265 Lot # WB322631) and goat-anti-rabbit 405 (Invitrogen Cat # A31556 Lot # 2249029) was used. Following this, slices were washed 3 more times, 10 min each wash, and subsequently mounted. To stain for CD206 and P2RY12 (Fig. 1a−k)-sections were blocked in 10% normal donkey serum, 1%BSA, and 2% Triton X-100 in 1× PBS. CD206 was applied at 1:100 and P2RY12 at 1:300 (Anaspec Cat. No AS55043A) overnight at 4 °C on an orbital shaker in blocking buffer without Triton-X 100. The next day, slices were washed with 1XPBS 3 times, with each wash lasting 10 min. Secondary antibodies were then applied in the same blocking buffer without Triton X-100 for 1 h at room temperature on an orbital shaker. Secondary antibodies were applied at 1:500 and included donkey-anti-rat 594 (Invitrogen Cat. No. A21209, Lot. No. 2041649) and donkey-anti-rabbit 647 (Invitrogen Cat. No. A31573 Lot No. 2420695). To stain for COX1, Iba1, and Aquaporin-4-sections were blocked in 10%normal goat serum, 1%BSA, and 2% Triton-X 100 in 1X PBS for 1 h at room temperature on an orbital shaker. eGFP was additionally stained for in Aldh1l1-eGFP mice using this blocking buffer. Sections were then washed 3 times in 1XPBS, where each wash was 10 min. COX1 was applied at 1:100 (ThermoFisher Scientific Cat. No. 35-8100), Aquaporin-4 at 1:250 (Sigma A5971), Iba1 at 1:100 (Wako Chemicals Cat. No. 018-28523), and eGFP at 1:250 (ThermoFisher Cat. No. A10262) overnight at 4 °C on an orbital shaker in blocking buffer without Triton-X 100. The next day, sections were washed with 1XPBS 3 times, with each wash lasting 10 min. Secondary antibodies were then applied at 1:500 in blocking buffer without Triton-X 100 for 1 h at room

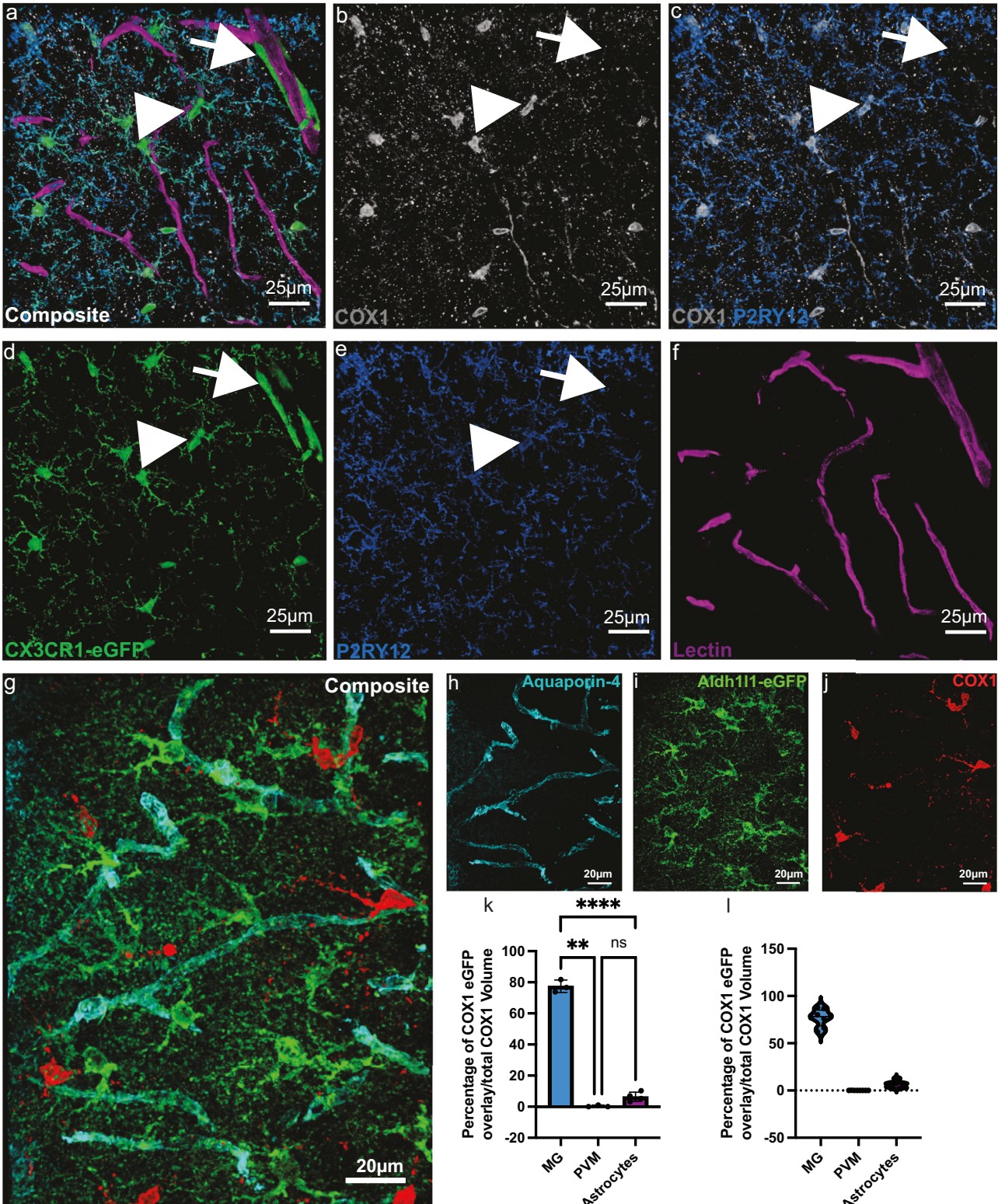

temperature. Secondaries included goat-anti-mouse 488 (Invitrogen Cat. No. A11001 Lot No. 2513496), goat anti-mouse 647 (Invitrogen Cat. No. A21235, Lot No. 2674387), goat-anti-rabbit 647 (Invitrogen Cat. No. A32733 Lot. No. YB36308), and goat-anti-chicken 488 (Invitrogen Cat. No. A11039 Lot No. 2566343). This same procedure was used for pericyte staining, which utilized CD13 (0.8 µg/mL, R&D Systems, # AF2335) and a retroorbital injection of DyLight 594-LEL (10 µL Vector Laboratories, # DL-1177) was performed to label blood vessels with

lectin. Goat anti-mouse 647 was used as a secondary antibody. For all stains involving a nuclear label, NucBlue Live Cell Stain ReadyProbes (Invitrogen Cat. No R37605 Lot No. 2272576) was applied at 2 drops per mL in blocking buffer without Triton X-100 for 30 min. This was the final step in all stains. Sections were then washed 3 times in 1× PBS, with each wash lasting 10 min. All sections were mounted with ProLong Glass Antifade Mountant (Invitrogen Cat. No. P36980 Lot. No. 2521164). Images were acquired with an Olympus FV10-MCPSU

**Fig. 6 | COX1 protein expression in microglia is higher than perivascular macrophages and astrocytes. a** 3D reconstruction of a 63X confocal image from somatosensory cortex in a CX3CR1-eGFP mouse. Myeloid cells are green, P2RY12 is blue, COX1 is white, and vessels are magenta. The arrow denotes a CX3CR1-eGFP[low] perivascular macrophage and arrowhead denotes a CX3CR1-eGFP[high] microglial cell. **b** 3D reconstruction of the same image in **a** but just showing the COX1 channel. **c** 3D reconstruction of the same image as in **a** but just showing the COX1 and P2RY12 channels. Note the arrow is in the same location as in **a** and confirms that the CX3CR1-eGFP myeloid cell there is P2RY12[negative]. **d** 3D reconstruction of the same image as in **a** but just showing the eGFP channel. **e** 3D reconstruction of the same image as in **a** but just showing the P2RY12 channel. **f** 3D reconstruction of the same image as in **a** but just showing the lectin channel. **g** 3D reconstruction of a 63X confocal image from somatosensory cortex in an Aldh1l1-eGFP mouse. Astrocytes are in green, the water channel Aquaporin-4 labeling astrocyte endfeet is in cyan, and COX1 is in red. **h** 3D reconstruction of the same image as in **g**, but just the Aquaporin-4 channel. **i** 3D reconstruction of the same image as in **g**, but just the Aldh1l1-eGFP channel. **j** 3D reconstruction of the same image as in **g**, but just the COX1 channel. **k** Bar graph quantifying the percentage of total COX1 volume that is overlaid with CX3CR1-eGFP or Aldh1l1-eGFP volume. $n = 8$ fields of view across 3 mice for measurements in CX3CR1-eGFP mice and 8 fields of view across 4 mice for measurements in Aldh1l1-eGFP mice. Brown-Forsythe ANOVA test with Dunnett's T3 multiple comparisons test. Multiplicity adjusted $p$ values- MG vs. PVM, $p < 0.0019$; MG vs. Astrocytes, $p < 0.0001$; PVM vs. Astrocytes, $p < 0.0688$. **l** Violin plot showing the raw data used to generate the bar graph in (**k**). Data presented as mean ± s.e.m. LUTs have been adjusted in shown images. Source data are provided as a source data file.

confocal microscope and 20× oil NA 0.85 objective, a Leica Stellaris 5, or Leica Stellaris 8 confocal microscope with 20× dry NA 0.75 and 63× oil NA 1.40 objectives. One to 3 images were acquired of each somatosensory cortical area in one hemisphere of one brain slice.

**Tamoxifen injections.** Tamoxifen was purchased from Sigma, cat. no. T5648. A 20 mg/mL solution was prepared by dissolving 200 mg first in 1 mL of 100% ethanol, vortexed, and then added to 9 mL of 37 °C pre-warmed corn oil. This was vortexed until fully dissolved. Mice were injected intraperitoneally once per day for 5 consecutive days at 100 mg/kg. On each day of injection, tamoxifen was placed at 37 °C and allowed to incubate all day before injecting mice (Fig. 7).

### Image analysis
**CD206± perivascular macrophage (PVM) and CX3CR1-eGFP± microglia vascular distribution analysis.** Images from CX3CR1-eGFP mice were first saved as TIFF files. These TIFF files were then opened in NIS-Elements (Nikon) and joined, appropriately labeled, and pixel to micron ratio set to that of the original image. Background subtraction was performed using the rolling ball radius method and a median filter kernel size 3 subsequently applied. A threshold was then applied so that all CD206 + PVM structures, eGFP+ cells, and lectin+ vessels were generated in a binary layer. Each individual CD206+ structure along α-smooth-muscle actin+ (αSMA) arterioles (>10 μm), αSMA-venules (greater than 10 μm), capillaries, or the pial layer were then individually selected and summed together to get total area for each vessel category. Each CD206+ vessel coverage area was then normalized to total vessel area. For arterioles and venules, total vessel area was determined by using the polygonal ROI tool and outlining the vessel area for each individual vessel, where lectin and CD206 PVM coverage was used to determine vessel boundaries. Total capillary area was determined by thresholding the lectin channel. The pial vasculature plus arteriole and venule labeling was subtracted. For eGFP+ cell coverage, a binary layer of eGFP+ structures was first generated. Then another intersection was generated between the first intersection and the lectin+ vasculature. This is the area reported for eGFP+ vascular coverage. Those cells contacting αSMA+ vascular smooth cell arterioles were summed for arteriole coverage, the same for αSMA- venule coverage, and capillary coverage (all vessels <10 μm).

**Perivascular macrophage and microglia CX3CR1-eGFP fluorescence intensity analysis.** Images were first opened in FIJI and individual channels separated and saved as a.TIFF file. These TIFF files were then opened in NIS-Elements (Nikon) and joined, appropriately labeled, and pixel to micron ratio set to that of the original image. P2RY12−/CD206+/CX3CR1-eGFP low perivascular macrophages were then identified. The autodetect ROI in Elements was then used to automatically draw an ROI around the perivascular macrophage eGFP signal and then intensity was measured. This procedure was repeated for the nearest P2RY12+/CD206−/CX3CR1-eGFP[high] microglia. A paired-end analysis was then used to statistically compare these side-by-side cells, as shown in Fig. 1k.

**Perivascular Macrophage, CX3CR1-eGFP±/Iba1± Microglia Cell Counting.** Images were analyzed in FIJI using the cell counter plugin and moving through the z-stack individual optical section by individual optical section. For perivascular macrophage cell counting (Supplementary Fig. 1m–o), eGFP[low] cells were identified and only counted if, at the appearance of a cell soma (largest circumferential portion), a nucleus was contained within the eGFP[low] structure. For CX3CR1-eGFP+/Iba1+ cell counting (Fig. 1a–e), the CX3CR1-eGFP channel first underwent cell counting, where the procedure for perivascular macrophage counting was repeated. Only nucleated cells were counted, and on this initial round of counting, the one performing the analysis was blind to the Iba1 channel. After recording total number of CX3CR1-eGFP cells, then a second count was performed with the Iba1 channel, and only those nucleated cells that were dual labeled were counted. The total number of CX3CR1-eGFP cells was then divided by the total number of Iba1 cells and multiplied by 100 to get the reported percentage.

**Capillary and arteriole diameter analysis.** To randomly generate stretches of capillary for diameter analysis following PLX3397 administration, a squared grid was applied to each image, such that 9 quadrants were evident. Google random number generator was then used to randomly generate a quadrant number. If capillaries were present in that region, google random number generator was then used again to generate an optical section number. If a stretch of capillary was present at this optical section number, a series of optical sections covering that stretch of capillary was selected, and surrounding fiduciary landmarks were used to determine the same start and stop location for each timepoint. A sum intensity projection was then created and Vasometrics[56] initiated to measure capillary diameter. The same parameters were used for each timepoint (Figs. 3, 7 and Supplementary Figs. 5, 9).

Arterioles were identified based on direction of flow and their lack of capillary branching relative to venules. This has been observed in our prior work following Alexa633 hydrazide administration[35]. At least two to three, ten optical section Vasometric measurements were performed for each arteriole, where the first measurement occurred right at the penetrating portion of the arteriole. These two to three measurements were then averaged to give one value for that arteriole. Arterioles per mouse were then averaged together to give a value for each mouse. This was repeated the same way at all timepoints and results analyzed using a paired end $t$-test. Vasometric measurements were acquired the same as that for the capillary diameter measurements.

**Capillary volume analysis.** Images were first opened in FIJI and individual channels separated and saved as a.TIFF file. These TIFF files were

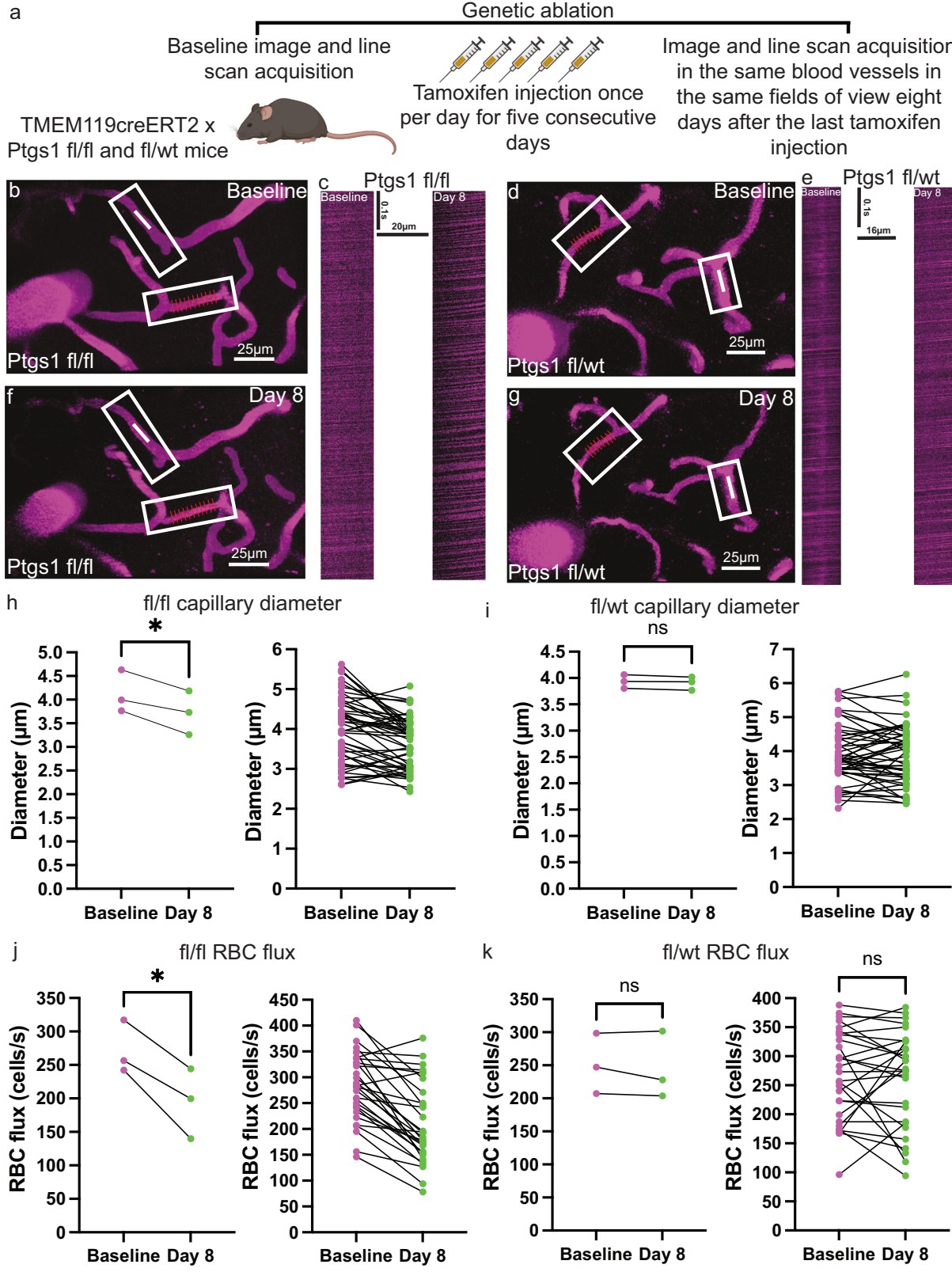

then opened in NIS-Elements (Nikon) and joined, appropriately labeled, and pixel to micron ratio set to that of the original image. This was done for both baseline and DPA8 images. A square ROI was then drawn around the capillary with ablated microglia in the baseline image. This capillary was identified based off the annotated z-stack from the timepoint of acquisition. That same square ROI was then drawn around the exact same capillary at the DPA8 timepoint. Then the exact same number of optical sections was selected to encompass the entire capillary region occupied by the ablated microglia. The images were duplicated based off the new z series. The rotating rectangle feature was then used to select out the capillary where ablation occurred. This generates a new image without affecting the underlying

**Fig. 7 | Genetic ablation of microglial COX1 results in significantly reduced capillary red blood cell flux and diameter. a** Cartoon depicting the experimental approach (*Created in BioRender. Eyo, U. (2025)* https://BioRender.com/h22a342). 3D reconstructions of a field-of-view at baseline (**b**) and 8 days after the last tamoxifen injection (**f**) in a TME119creERT2 Ptgs1fl/fl mouse. The capillary with red lines in the bottom white box represent the vasometrics output measurement for diameter. The top white box indicates the capillary from which a line scan was acquired, as shown in (**c**). **c** line scan at baseline (left) and day 8 after the last tamoxifen injection (right). 3D reconstructions of a field-of-view at baseline (**d**) and 8 days after the last tamoxifen injection (**g**) in a TME119creERT2 Ptgs1fl/wt mouse. The capillary with red lines in the top white box represent the vasometrics output measurement for diameter. The bottom right white box indicates the capillary from which a line scan

was acquired, as shown in (**e**). **e** line scan at baseline (left) and day 8 after the last tamoxifen injection (right). **h** before-and-after plot showing the averaged capillary diameter values (left) and raw values (right) at baseline and day 8 in Ptgs1 fl/fl mice. $n = 51$ capillaries/3 mice. Two-tailed paired $t$-test, $p < 0.0318$. **i** Before-and-after plot showing the averaged capillary diameter values (left) and raw values (right) at baseline and day 8 in Ptgs1 fl/wt mice. $n = 51$ capillaries/3 mice. Two-tailed paired $t$-test, $p < 0.1050$. **j** before-and-after plots showing the averaged RBC flux values (left) and raw values (right) at baseline and day 8 in Ptgs1 fl/fl mice. $n = 31$ capillaries/3 mice. Two-tailed paired $t$-test, $p < 0.0287$. **k** before-and-after plots showing averaged values (left) and raw values (right) of RBC flux from control Ptgs1 fl/wt mice. $n = 29$ capillaries/3 mice. Two-tailed paired $t$-test, $p < 0.7074$. Source data are provided as a source data file.

metadata. Finally, images underwent a rolling ball background subtraction, median filtered with kernel size 3, and then a threshold applied to both images- using the exact same thresholding parameters. Volume of capillary was then recorded. This procedure was then repeated for the neighboring capillary segment where ablation did not occur (Fig. 5).

**CD206 and CX3CR1-eGFP high volume analysis following PLX3397 administration and withdrawal.** Images were first opened in FIJI and individual channels separated and saved as a.TIFF file. These TIFF files were then opened in NIS-Elements (Nikon) and joined, appropriately labeled, and pixel to micron ratio set to that of the original image. All images then underwent a rolling ball background subtraction, median filter kernel size 3 application, and threshold applied to generate binary CD206 and CX3CR1-eGFP$^{high}$ layers. The same threshold was applied at all timepoints between images, and volume of each layer recorded. Note, that the recorded perivascular macrophage volume reported was generated by selecting all structures in the binary CD206 layer that were located in parenchymal space and adding their volumes together. The meningeal macrophage volume reported was generated by selecting all structures in the binary CD206 layer at the pial surface or top of the slice and adding their volumes together. Any eGFP structures located at the top of the slice or pial layer were removed before recording CX3CR1-eGFP$^{high}$ volume so that we reported only the volume of structures in parenchymal space.

**COX1 volumetric analysis in astrocytes, perivascular macrophages, and microglia.** Images were first opened in FIJI and individual channels separated and saved as a.TIFF file. These TIFF files were then opened in NIS-Elements (Nikon) and joined, appropriately labeled, and pixel to micron ratio set to that of the original image. Images then underwent a rolling ball background subtraction, median filter kernel size 3, and threshold set. This was done for both the COX1 channel and eGFP channel. Then, a new binary layer was generated between the eGFP and COX1 binary layers using the Boolean AND operator in Elements. Then, the eGFP/COX1 overlay volume was recorded, and finally total COX1 volume recorded. This is how astrocyte and microglia COX1 expression was determined. Because perivascular macrophages are CX3CR1-eGFP$^{low}$, the rotating rectangle feature was used to isolate them from the rest of the image. To generate a binary layer, the image underwent rolling ball background subtraction, median filter kernel size 3 application, and then subsequent threshold application. Measurements were recorded the same as for microglia and astrocytes. It is important to note that the threshold applied for the eGFP layers was slightly different between the three cell types, so it is possible that the raw data is not fully accurate. However, given the obvious expression of COX1 in microglia as assayed by immunohistochemistry, we have no doubt that our conclusion is correct (please see the images in Fig. 6). In each instance, the minimum threshold needed to create a binary layer of entire cells was applied. The graph in Fig. 6 is reporting the volume of the eGFP/COX1 layer divided by the total COX1 volume of that cell and expressed as a percentage.

**Microglia area analysis following surgery.** Images from all time-points were opened in FIJI and a sum intensity projection using the exact same number of optical sections was created. One ROI large enough to crop out individual microglia was made and saved to the ROI tool. This was then applied to individual microglia across all timepoints. Cropped images then underwent background subtraction, a median filter of kernel size three, and then threshold set. The area was then measured from the generated binary layer and recorded.

**Manual red blood cell flux line scan analysis.** Every line scan (at least 60 s long) acquired at baseline and a subsequent time point was opened in FIJI and duplicated. Then, the pixel size of a 1 s window was calculated and the line scan cropped at the exact same starting point for both images. We then manually counted the number of red blood cell streaks occurring in that one second window. Some streaks did appear thicker, and as previously reported[57], likely corresponded to a rouleaux of RBCs. However, we felt we could identify individual cells in these instances, and they were counted as such if a clear line of rhodamine or TRITC dye separated them. If there was the appearance of stalling in the line scan, it was not selected for a one second window flux analysis, thus ensuring we are not actually comparing normal flow to a stall in any of our experiments (Figs. 3 and 7). The raw number at baseline was then compared to the raw number at the subsequent timepoint following experimental intervention (Figs. 4 and 7).

**Microglia-capillary "hotspot" analysis.** Two photon images of the same field of view before, at the end of PLX3397 treatment, and after PLX3397 withdrawal were analyzed. Capillary regions with associated microglia were marked in the before images and transposed to the after images to determine whether repopulated microglia were within 5 μm of the initial "hotspot".

**CAM vs. non-CAM COX1 expression and COX1 expression in Ptgs1 fl/wt and fl/fl mice.** Images were first opened in FIJI and individual channels separated and saved as a.TIFF file. These TIFF files were then opened in NIS-Elements (Nikon) and joined, appropriately labeled, and pixel to micron ratio set to that of the original image. Images then underwent a rolling ball background subtraction, median filter kernel size 3, and threshold set. A CAM was identified along with its closest non-CAM neighbor. The z series corresponding to these cells were then selected and the rotating rectangle feature then used to isolate each cell. The volume for COX1 and eGFP layers (Supplementary Fig. 7) or Iba1 and COX1 layers (Supplementary Fig. 8) were recorded following image segmentation. The CAM versus nonCAM values were then compared using a two-tailed paired end $t$-test. The Ptgs1 fl/wt and fl/fl values were compared using a two-tailed unpaired $t$-test.

**RNAseq**
**Data acquisition and processing.** The RNA sequencing dataset was obtained from the Allen Brain Map portal (https://portal.brain-map.org/atlases-and-data/rnaseq, specifically https://data.nemoarchive.

org/biccn/grant/u19_zeng/zeng/transcriptome/scell/10x_v2/mouse/processed/analysis/10X_cells_v2_AIBS/). The dataset includes gene expression matrices and metadata files containing barcode identifiers, feature annotations, and cell type classifications.

**Preprocessing and quality control.** The raw data files, including barcodes (barcodes.tsv.gz), gene features (features.tsv.gz), and expression matrix (matrix.mtx.gz), were loaded into R using the Seurat, Matrix, and ggplot2 packages. Cell annotations were imported from cluster.membership.csv and cluster.annotation.csv, which were merged to obtain cell type classifications. To identify glial cells, a subset of cell types including astrocytes, macrophages, oligodendrocytes, oligodendrocyte precursor cells (OPCs), and endothelial cells was extracted using pattern matching (grep). The resulting dataset was saved as glial_cells.csv.

**Barcode cleaning and matching.** Barcodes from the expression matrix were trimmed to remove leading commas and quotes, ensuring consistency between expression data and metadata. A matching process was performed to align glial cell barcodes with those in the expression matrix using the match function.

**Gene expression extraction.** To assess *Cox1* (*Ptgs1*) expression in glial cells, the row corresponding to *Ptgs1* in the feature annotation file was identified using grep. The expression values were extracted from the filtered expression matrix and converted into a numeric vector. Data consistency was verified by ensuring that the number of glial cells matched the length of the extracted expression vector.

**Visualization and data representation.** A bar plot was generated to visualize *Ptgs1* expression across glial cell types using ggplot2. The subclass_label column was modified to rename "Macrophage" to "Microglia/Macrophage" for clarity. Factor levels were reordered to present microglia/macrophages first, followed by endothelial cells and astrocytes. The final plot displayed *Ptgs1* expression levels (UMI) across these cell types, with customized colors assigned to each category. All data preprocessing and analysis steps were performed using R version 4.3.1 with the specified packages.

### Statistics
GraphPad Prism software was used to perform all statistical analyses. Details for every statistical test are reported in the figure legends. Every parametric test used was validated by first performing tests of normality. Parametric tests were further selected based on datasets having equivalent or different standard deviations, where a difference of <1.5 was counted as being equal. All error bars in bar graphs represent the standard error of the mean (s.e.m).

### Reporting summary
Further information on research design is available in the Nature Portfolio Reporting Summary linked to this article.

## Data availability
All data are available from the corresponding authors on request. The RNA sequencing dataset was obtained from the Allen Brain Map portal (https://data.nemoarchive.org/biccn/grant/u19_zeng/zeng/transcriptome/scell/10x_v2/mouse/processed/analysis/10X_cells_v2_AIBS/). Source data are provided with this paper.

## Code availability
The code used for sequencing analysis in this paper has been uploaded to the Eyo lab website (https://www.microgleyolab.com/s-projects-basic).

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

## Acknowledgements

We thank members of the Eyo lab (especially Dr. Hongbin Li), the Center for Brain Immunology and Glia (BIG) and the Cardiovascular Research Center (CVRC) for valuable discussions in the development of this project. We additionally thank Dr. Maude Bordeleau for providing the mouse samples used in the electron microscopy image. This study received the following support: National Institute of Neurological Disorders and Stroke, NIH NS122782 and NIH NS119243 (U.B.E). National Heart, Lung, and Blood Institute- NIH HL007284 (W.A.M), NIH HL137112 and NIH HL171997 (B.E.I), Owen's Family Foundation (U.B.E), UVA Brain Institute Postdoctoral Fellowship (W.A.M.), American Heart Association Postdoctoral Fellowship (W.A.M) (25POST1376070, accessible at https://doi.org/10.58275/AHA.25POST1376070.pc.gr.227309), M.È.T is Canada Research Chair (Tier II) in *Neurobiology of Aging and Cognition*. F.G.I was supported by a doctoral scholarship from the Mexican Council of Science and Technology (CONAHCYT) and Health Research BC, Michael Smith fellowship.

## Author contributions

W.A.M. III and U.B.E. conceived of and developed the project. N.A.S., A.O.L.-O., D.H.L., F.G.I., P.A., E.R., A.G., D.G., and A.S. contributed analysis to the project. M.-E.T. and F.G.I. provided EM analysis. B.E.I., and U.B.E. contributed equipment and resources for the study. W.A.M. III, and U.B.E. contributed reagents and resources for the project. W.A.M. III, and U.B.E. contributed to the writing of the manuscript. W.A.M. III, B.E.I., and U.B.E. contributed to the editing of the manuscript. U.B.E. oversaw the project.

## Competing interests

The authors declare no competing interests
