## [Transparent Peer Review file · Nature Communications]

Microglial cyclooxygenase-1 modulates cerebral capillary basal tone in vivo

Corresponding Author: Dr Ukpong Eyo

Version 0:

Reviewer comments:

Reviewer #1

(Remarks to the Author)

The reviewers raised several concerns about the manuscript, focusing mainly on the lack of conceptual novelty, the specificity of cell types targeted, the interpretation of the data in the context of microglial versus macrophage function, and the relevance of the findings to human disease and potential therapies. Addressing the last issue might substantially elevate the novelty and clinical relevance of findings distinguishing it from recently published manuscripts on capillary-associated microglia and capillary constriction.

1. One of the issues was the lack of human data showing capillary constriction in pathological conditions. While the authors acknowledged that they didn't generate such data themselves, they addressed this point by citing five relevant papers where capillary constriction has been documented in human studies. This is a reasonable way to handle the concern without doing new experiments, though the addition of even limited human data would have strengthened the manuscript.

2. The more serious concerns came in the form of conceptual and experimental discrepancies. First, the reviewer questioned the specificity of the ablation approaches, particularly whether the authors were really targeting capillary-associated macrophages (CAMs) or just broadly eliminating microglia. The authors pushed back here, suggesting that the term "capillary-associated macrophages" might be misleading, since their data show that fewer than 1% of macrophages associate with capillaries. They further clarified that they only used one ablation method for macrophages (PLX) and multiple for microglia (including 2Phatal and a COX1 conditional knockout). However, this response didn't really resolve the concern. Instead it sidestepped the request for more specific targeting. Instead of refining the identity of the targeted cells, the authors debated the terminology.

The 2Phatal experiments were another point for the reviewer's question who wanted to know exactly what types of cells were ablated. The authors explained that the ablated cells are CX3CR1-high, meaning they are microglia rather than BAMs, which tend to express CX3CR1 at lower levels. The authors didn't show TMEM119 staining in vivo (which would have provided more definitive identification), citing technical challenges with imaging.

Instead, they argued that it's reasonable to assume these are TMEM119-positive microglia: "While we do not know about whether these are Tmem119-positive, because they are microglia, they are almost certainly Tmem119-positive as well." Such assumption is not sufficient. While this is a fairly standard interpretation given the field's reliance on CX3CR1 and TMEM119 as markers. TMEM119 is known to be more specific for microglia and CX3CR1 is expressed in both microglia and BAMs, though at different levels. However, TMEM119 expression varies substantially in different areas of the brain and in many pathological situations it is quite low. The main relevance of the present study is that capillary constriction is critical in pathologies. Thus, there is a need to show that the capillary-associated cells that are targeted in ablation experiment express sufficient level of TMEM119 and represent the same cells targeted in microglia-specific knockout.

3. Due to this same reason the main mechanistic claim that microglial COX1 regulates capillary tone is weakened since characterization and targeting strategies relied on different markers. In the characterization studies, the authors used CX3CR1-eGFP mice to identify microglia, but in the functional studies, they used TMEM119-Cre to knock out COX1. The authors argued that TMEM119-Cre provides the microglia specificity while the use of CX3CR1-Cre would hit BAMs as well. While this is valid and aligns with the literature, the reviewer's point remains. Without comparing the results with two Cre drivers, it's hard to rule out contributions from other cell types. The authors acknowledged this and said they're setting up those experiments now, but didn't include them in the current paper.

4. Another key point the reviewer made was that the effect of microglial COX1 loss on capillary tone in a disease model

(such as 5XFAD) wasn't shown. Addition of these data would remedy the problem with novelty of this study. The authors agreed this is a limitation and said they are beginning to perform those studies, but again framed it as outside the current scope.

Similarly, when asked about the therapeutic relevance of targeting COX, given COX1's known role in thrombosis and bleeding, the authors stated that more work is needed to understand how COX1 functions in pathology and how its role may differ depending on the disease stage or cell type.

5. One place where the authors added important new data was showing that TMEM119-driven COX1 knockout led to reduced COX1 expression in Iba1+ cells (The more relevant data should actually show reduced COX-1 expression in TMEM119-positive cells since TMEM119 was used as a Cre driver). They included this as a new extended figure, showing about a 50% reduction at the cell level, though statistical significance was limited when analyzed by animal (n = 3). This helps to support their argument, but the sample size is admittedly small.

6. Again, for some reasons in "Merged File containing manuscript text and 7 Figure files" the figures are not numbered and figure legends are missing, although these are present in a separate file of "Figures and Figure legends".

Please find our point-by-point responses to the Reviewer's comments below:

Reviewer #1 (Remarks to the Author):

The reviewers raised several concerns about the manuscript, focusing mainly on the lack of conceptual novelty, the specificity of cell types targeted, the interpretation of the data in the context of microglial versus macrophage function, and the relevance of the findings to human disease and potential therapies. Addressing the last issue might substantially elevate the novelty and clinical relevance of findings distinguishing it from recently published manuscripts on capillary-associated microglia and capillary constriction.

1. One of the issues was the lack of human data showing capillary constriction in pathological conditions. While the authors acknowledged that they didn't generate such data themselves, they addressed this point by citing five relevant papers where capillary constriction has been documented in human studies. This is a reasonable way to handle the concern without doing new experiments, though the addition of even limited human data would have strengthened the manuscript.

Response: We generally agree with and appreciate this summary and acknowledgment.

2. The more serious concerns came in the form of conceptual and experimental discrepancies. First, the reviewer questioned the specificity of the ablation approaches, particularly whether the authors were really targeting capillary-associated macrophages (CAMs) or just broadly eliminating microglia. The authors pushed back here, suggesting that the term "capillary-associated macrophages" might be misleading, since their data show that fewer than 1% of macrophages associate with capillaries. They further clarified that they only used one ablation method for macrophages (PLX) and multiple for microglia (including 2Phatal and a COX1 conditional knockout).

However, this response didn't really resolve the concern. Instead it sidestepped the request for more specific targeting. Instead of refining the identity of the targeted cells, the authors debated the terminology.

The 2Phatal experiments were another point for the reviewer's question who wanted to know exactly what types of cells were ablated. The authors explained that the ablated cells are CX3CR1-high, meaning they are microglia rather than BAMs, which tend to express CX3CR1 at lower levels. The authors didn't show TMEM119 staining in vivo (which would have provided more definitive identification), citing technical challenges with imaging.

Response: The last point is a fair and welcomed critique that we didn't show TMEM119 staining in vivo. We have made this acknowledgment as a limitation in point # 2 above.

Instead, they argued that it's reasonable to assume these are TMEM119-positive microglia: "While we do not know about whether these are Tmem119-positive, because they are microglia, they are almost certainly Tmem119-positive as well."

Such assumption is not sufficient. While this is a fairly standard interpretation given the field's reliance on CX3CR1 and TMEM119 as markers. TMEM119 is known to be more specific for microglia and CX3CR1 is expressed in both microglia and BAMs, though at different levels. However, TMEM119 expression varies substantially in different areas of the brain and in many pathological situations it is quite low. The main relevance of the present study is that capillary constriction is critical in pathologies. Thus, there is a need to show that the capillary-associated cells that are targeted in ablation experiment express sufficient level of TMEM119 and represent the same cells targeted in microglia-specific knockout.

Response: We appreciate that we are recognized for following an interpretation of the field that is fairly standard. While we agree that TMEM119 varies in expression by brain regions and pathology, we are not (in this study) varying the brain region (since we are looking in one region: the somatosensory cortex) and we are not looking in any pathology. Finally, we do not accept the claim that the "main relevance of the present study is that capillary constriction is critical in pathologies". This statement is not congruent with the main thought of the current version of our manuscript. Specifically, we situate our study in basal physiology and NOT pathology which is why we felt it reasonable to take out the previous 5XFAD data.

3. Due to this same reason the main mechanistic claim that microglial COX1 regulates capillary tone is weakened since characterization and targeting strategies relied on different markers. In the characterization studies, the authors used CX3CR1-eGFP mice to identify microglia, but in the functional studies, they used

TMEM119-Cre to knock out COX1.

The authors argued that TMEM119-Cre provides the microglia specificity while the use of CX3CR1-Cre would hit BAMs as well. While this is valid and aligns with the literature, the reviewer's point remains. Without comparing the results with two Cre drivers, it's hard to rule out contributions from other cell types. The authors acknowledged this and said they're setting up those experiments now, but didn't include them in the current paper.

Response: We appreciate this reviewer's acknowledgement that our point "is valid and aligns with the literature". Our goal was to rule in microglia at capillaries (and not necessarily rule out other cells). We had included this point for astrocytes but have now included it for BAMs as well in the last paragraph of our manuscript as a limitation as mentioned in point #1 above.

4. Another key point the reviewer made was that the effect of microglial COX1 loss on capillary tone in a disease model (such as 5XFAD) wasn't shown. Addition of these data would remedy the problem with novelty of this study. The authors agreed this is a limitation and said they are beginning to perform those studies, but again framed it as outside the current scope.

Similarly, when asked about the therapeutic relevance of targeting COX, given COX1's known role in thrombosis and bleeding, the authors stated that more work is needed to understand how COX1 functions in pathology and how its role may differ depending on the disease stage or cell type.

Response: We acknowledge this summary.

5. One place where the authors added important new data was showing that TMEM119-driven COX1 knockout led to reduced COX1 expression in Iba1+ cells (The more relevant data should actually show reduced COX-1 expression in TMEM119-positive cells since TMEM119 was used as a Cre driver). They included this as a new extended figure, showing about a 50% reduction at the cell level, though statistical significance was limited when analyzed by animal (n = 3). This helps to support their argument, but the sample size is admittedly small.

Response: This is a useful critique and we acknowledge our limitation in not labelling Tmem119 in the COX1 KO cells and the smaller sample size as point #3 of the limitation above

6. Again, for some reasons in "Merged File containing manuscript text and 7 Figure files" the figures are not numbered and figure legends are missing, although these are present in a separate file of "Figures and Figure legends".

Response: We again apologize for this but are glad they could get the necessary files.